# Spatiotemporal Characteristics of Coupling and Coordination of Cultural Tourism and Objective Well-Being in Western China

**DOI:** 10.3390/ijerph20010650

**Published:** 2022-12-30

**Authors:** Lili Pu, Xingpeng Chen, Li Jiang, Hang Zhang

**Affiliations:** 1College of Earth and Environmental Sciences, Lanzhou University, Lanzhou 730000, China; 2Institute of County Economic Development & Rural Revitalization Strategy, Lanzhou University, Lanzhou 730000, China; 3Tourism College, Qinghai Normal University, Xining 810000, China

**Keywords:** cultural tourism–objective well-being (CT–OWB), coupling and coordination, evaluation index system, western China, sustainable development

## Abstract

A supply of high-quality cultural tourism products effectively promotes people’s happiness. This study takes the coupling and coordination relationship between cultural tourism (CT) and objective well-being (OWB) in western China as the research object and constructs an index evaluation system for the development levels of cultural tourism and objective welfare, which are divided into three development stages of dysfunctional, transitional, and coordinated development and 10 coordination levels, including high-quality coordination. The entropy weight method, coupled coordination model, Thiel index, standard deviation, and coefficient of variation were used to calculate the comprehensive development index and coupling coordination degree of the CT and OWB systems in western China from 2007 to 2020, and then their evolution characteristics were analyzed from the perspectives of time and space. The results show the following: (1) The comprehensive development capacity of the CT and OWB systems in the western region shows a relatively consistent growth trend except for 2020, the overall development capacity of objective welfare was relatively high, and the development capacity of cultural tourism was relatively low. (2) The CT and OWB systems were in a state of transition from high coupling and low coordination to high coupling and high coordination, which were divided into three development stages: imbalanced stage (2007–2008), transitional stage (2009–2011), and coordinated development (2012–2020). The coordination degree has developed steadily from moderate misalignment to good coordination over time, and the diversified characteristics of coupling coordination levels are obvious. (3) The spatial equilibrium of the CT and OWB systems is obvious, and the spatial difference in the western region is getting smaller, but the relative gap is widening significantly. (4) The comprehensive development capacity of the cultural tourism system should be enhanced, the spiritual needs of residents based on objective well-being should be met, and the sustainable development of the CT and OWB systems.

## 1. Introduction

With the improvement of China’s economic and social levels and the continuous improvement of per capita consumption capacity, cultural tourism has become a trend of industrial development [1] and a necessity in people’s lives [2]. In this continuous development trend, China will enter an important era of development with cultural tourism resources at the core [3]. In particular, the establishment of the Ministry of Culture and Tourism of China in 2018 broke the institutional bottleneck of cultural tourism development. How to give full play to the high-quality development of cultural tourism has become an urgent problem to be solved in terms of economic and social development, and new cultural tourism products such as the metaverse [4], cloud tourism [5], and digital tourism [6] have become new research hot spots and development trends, helping the life of cultural tourism and development [7]. The continuous emergence of new business formats is conducive to giving full play to the comprehensive benefits of cultural tourism [8] and continuously promoting the high-quality development of the regional economy and society. Cultural tourism is an important way to meet people’s material and spiritual needs [9], which is conducive to improving their sense of happiness and narrowing the gaps between groups in spiritual and cultural life [10]. In particular, realizing the value of cultural tourism capital and reconstructing cultural tourism forms, integrating excellent traditional culture and production and life, and facilitating residents’ well-being [11] can better ensure the mutual development of the CT and OWB systems. It can be seen that the coordinated development of the two systems is conducive to sustainable economic and social development and the realization of people’s happiness.

However, at present, the development of cultural tourism is facing problems such as abundant resources but relatively backward economic and social development [12] and the development opportunities of cultural tourism integration under the background of digital economy development [13], but China’s cultural tourism development is characterized by abundant resources and strong economic strength [14]. How to realize the coupling and coordinated development of the two systems of cultural tourism and objective welfare has become an important driving force for the sustainable development of the regional economy and also an important measure for the continuous improvement of the comprehensive benefits of cultural tourism. Therefore, taking the coupling and coordinated development of the two systems of cultural tourism and objective well-being as the starting point, this paper constructs a coupling and coordination model of the two systems and conducts empirical analysis in western China.

First, relying on the actual situation of China’s development and industrial characteristics, we build a comprehensive evaluation index system of two systems of cultural tourism and objective welfare, including a total of 6 secondary indicators and 18 evaluation factors, and then we build a coupling and coordination model of the two systems of cultural tourism and objective welfare, which is divided into 3 development stages and 10 coordination levels. Taking western China as an example, this paper provides a typical case of the coupling and coordinated development law and its characteristics of the two systems of cultural tourism and objective welfare in China and enriches the relevant theories of the interactive development of cultural tourism and objective welfare.

Second, through the overall construction of the cultural tourism product system of regional destinations, we build a more comfortable, safe, and convenient objective welfare system and provide China’s typical cases and sustainable development paths for the coupling and coordination of high-quality development of cultural tourism and objective welfare.

In 1985, the World Tourism Organization (UNWTO) defined the conceptual meaning of cultural tourism from both broad and narrow levels, and in 1991 the European Association for Tourism and Leisure Education (ATLAS) defined cultural tourism as “all non-profit activities in the location of cultural resources such as ancient ruins, artistic and cultural performances, and artistic interpretations in order to obtain or meet their own cultural needs”. The 1999 International Charter on Cultural Tourism elaborates on the meaning of cultural tourism, which regards culture itself and the environment as the core of cultural tourism, and the cultural environment includes not only historical and cultural landscape resources, but also local residents’ customs, lifestyles, and even the natural landscape resources of tourist destinations. Tourism and culture go hand in hand and complement each other [15], cultural communication plays an important role in the development of tourism economy [16], and likewise the process of tourism presents diverse cultural needs [17]. With the development of the economy and society, the concept of cultural tourism is also constantly changing, tourism has become a necessity of people’s lives, and more attention is being paid to the experience of regional culture. Therefore, the article understands cultural tourism as a way of people’s modern lives. Objective well-being mainly represents people’s happiness and spiritual happiness in life [18], and both of them are highly correlated in spiritual life, because this paper takes western China as an example to analyze the coupling and coordination relationship between cultural tourism and objective well-being to further promote the sustainable development of the region.

Cultural tourism and well-being have been a hot topic and a focus of academic discussion [19,20,21], and cultural tourism is part of the tourism product [22]. With the increasing competitiveness and sustainability of the cultural tourism industry itself [23], the important role of economic and social development is increasingly emerging [24]. Because of its high degree of relevance, wide coverage, strong radiation, and driving force [25], recent studies have also paid more attention to population-oriented research [26], which plays an important role in promoting the well-being of residents and can solve social problems such as aging [27] and the healthy growth of children [28], along with other problems in social development, and has become a hot spot in academia [29].

Residents’ well-being mainly includes two parts: subjective well-being (SWB), and objective well-being (OWB). SWB occurs mainly through travel experiences and tourism activities, which tend to produce positive emotions [30], especially in psychological resources, leisure, and family life [31]. Similarly, organizational trust and recognition [32] and high levels of psychological capital [33] in the lodging industry promote employee well-being and contribute to well-being at work. Research on the OWB of residents mainly focuses on the relationship between cultural ecosystem service value [34,35], medical treatment [36], and leisure transportation [37] in nature reserves. The study of OWB is mainly based on external indicators that reflect residents’ objective economic and social living conditions, such as relative income [38,39], expenditure structure [40], employment status [41], physical health indicators [42], social security [43], and other aspects. Among other factors, education, entertainment, aesthetics, and escapism promote well-being [44], mainly in the process of tourism consumption [45], which can better promote the quality of life and happiness of residents. In the process of social development, consideration should be given to the application of OWB, such as by paying more attention to the well-being of residents in the process of tourism planning [46], and constantly optimizing the functions of the cultural tourism system in economic, social, and ecological aspects [47] to improve service efficiency. Improving residents’ perceptions of the impact of tourism has a clear effect on community, environmental, and economic benefits [48] and can enhance well-being [49,50,51], driving the development of common prosperity in China [10].

The effect of the development of cultural tourism on residents’ OWB has become prominent, and similarly, residents’ demands for cultural tourism has also become prominent; however, the academic community has focused less on the coupling and coordination between cultural tourism (CT) and OWB. Mainly, the research has been on a single culture or tourism industry and subjective welfare, while research on OWB is relatively lacking. Therefore, based on the relationship between CT and OWB, in this paper we took 12 provinces (regions) in western China as examples and selected statistical data from 2007 to 2020 to quantify the temporal and spatial characteristics of the coupling and coordination between CT and OWB. This study enriches the theoretical research on CT and the OWB of residents by providing typical case studies in China.

## 2. Study Area and Data Resources

### 2.1. Study Area

Western China mainly includes the 12 provinces (regions) of Guangxi, Gansu, Guizhou, Sichuan, Tibet, Shaanxi, Chongqing, Qinghai, Ningxia, Xinjiang, Yunnan, and Inner Mongolia, with the roof of the world, the Himalayas, plateau lakes, Gobi Desert, Loess Plateau, Qilian Glacier, Yellow River, Karst landforms, Yangtze River Three Gorges, and other natural landscapes, which are important ecological reserves. The region is also home to the Terracotta Army of the First Emperor of Qin, the Dunhuang Mogao Grottoes, the ruins of the Great Wall, the Mausoleum of the Yellow Emperor, the ancient Silk Road, the remains of ancient civilizations, the ruins of Yuanmouren, the Potala Palace, the Jokhang Temple, and other cultural landscapes. It is an important birthplace of Chinese civilization. By the end of 2020, the 12 provinces (regions) in the west had 97 5A-level tourist attractions, 3281 star-rated hotels, 9191 travel agencies, 4043 art performance groups, 1632 museums, and 1218 cultural centers, with a total of 5915.54 million tourist trips and tourism revenue of USD 568.09 billion (see a and b in Figure 1 for details). As of 2020, the number of mass cultural institutions in the western region was 16,981, employing 68,145 people, with an average of four employees at each cultural institution (see c and d in Figure 1 for details).

The role of cultural institutions in the cultural life of the masses is increasingly prominent, and creative development ideas are growing. Silk Road Flower Rain, Impression Liu Sanjie, Ma Tao Feiyan, Lijiang Ancient City, and other performing arts, cultural, and creative products for tourists (heritage tourism) have sprung up and gradually become necessities in people’s spiritual lives and the lifestyles that they enjoy. In particular, Yunnan Province, Guangxi Zhuang Autonomous Region, Guizhou Province, Sichuan Province, and other provinces and regions play important roles in the development of cultural tourism, such as eco-tourism, performing arts tourism, ethnic tourism, rural tourism, and heritage tourism in the west. Their unique resources are the cornerstones of the high-quality development of cultural tourism in the western region and will certainly bring a better life and future to people of all ethnic groups [52].

### 2.2. Building an Indicator System and Data Sources

According to the particularities of China’s economic development, the existing cultural tourism research [53], and the particularities of industrial development [54], in order to build a holistic cultural tourism system based on supply and demand, 5A-level tourist attractions, star-rated hotels, travel agencies, museums, cultural centers, and art performance groups should be considered as important parts of the system, playing an integral role in the quality of life of residents and tourists’ perceptions of happiness [55,56,57,58]. In particular, there is an aspect that cannot be ignored if we mainly consider the objective well-being of residents, and it is measured through income, consumption, and health. In the 1960s and 1970s, the United Nations Millennium Eco-System Assessment divided human well-being into five components: basic goods and services, security, health, social relations, and choices [30]. Based on data availability and representativeness in the available research [59,60], in this study we selected the four aspects of consumption, health, environment, and safety to construct an OWB system (Table 1).

The numbers of 5A tourist attractions, star tourist hotels, travel agencies, artistic performance groups, museums, and cultural centers, per capita tourism consumption of domestic tourists, per capita tourism consumption of inbound tourists, total library circulation, total visitors to museums, domestic art performance attendance, and cultural market institutions operating income are derived from the *Zhongguo Wenhua Wenwu He Lvyou Tongji Nianjian*. Per capita disposable income, per capita public green space area, public security expenditure, and traffic accident death and injury rates are derived from the *China Statistical Yearbook*. Traffic accident death and injury rates are calculated by the proportion of deaths and injuries to the number of traffic accidents. The number of beds in health institutions per 1000 population comes from the *China Health Statistics Yearbook (China Statistical Yearbook)*. Healthcare personnel per 1000 population are derived from the *China’s Tertiary Industry Statistical Yearbook*.

The western region is located in the western region of China (see Figure 2 for details) and plays an important role in China’s cultural tourism, ethnic tourism, eco-tourism, and rural tourism. As of the end of 2020, there were 4625 A-grade tourist attractions, accounting for 34.85% of the national A-level tourist attractions. From the perspective of natural resources, the western region accounts for 72% of the country’s land area, and the terrain falls from the roof of the world to the low-altitude plain, and the climate is distributed vertically. Since the policy support of reform and development and the large-scale development of the western region, the development of tourism in the western region has also shown a rapid growth trend, and tourism hotspots have emerged, such as Dunhuang Mogao Grottoes, Qinghai Chaka Salt Lake, Qin Shihuang Terracotta Army, Shapotou, and other important tourist scenic spots, and “cultural and tourism integration”, “cultural tourism”, and other hot spots have become hot words. The development of the cultural tourism industry in the western region mainly relies on important cultural routes such as the traditional Silk Road and the Tibet-Yi corridor to attract domestic and foreign tourists, and the comprehensive role of tourism in economic development, social construction, and foreign exchanges has been continuously been brought into play (see Table 2).

## 3. Methods

### 3.1. Entropy Weight Method

The index weights of the evaluation factors in the CT and OWB systems of 12 western provinces (regions) were determined by the entropy method [61], the specific calculation of indicator weights is shown in Figure 3, and the specific calculation formula was as follows:(1)Standardized processing of indicators:

Normalization of positive indicators:(1)$${x^{\prime }}_{\theta ij}=x_{\theta ij}/x_{max}$$

Normalization of negative indicators:(2)$${x^{\prime }}_{\theta ij}=x_{max}/x_{\theta ij}$$

(2)Determine the metric weight:


(3)
$$y_{\theta ij}={x^{\prime }}_{\theta ij}/\underset{\theta }{\sum }\underset{i}{\sum }{x^{\prime }}_{\theta ij}$$


(3)Calculate the entropy value of *j*:


(4)
$$e_{j}=-\;k\underset{e}{\sum }\underset{i}{\sum }y_{\mathrm{ij}}\ln(y_{\theta \mathrm{ij}})$$


*k* = ln(rn).

(4)Calculate the information utility value of item *j*:


(5)
$$g_{j}=1-e_{j}$$


(5)Calculate the weight of each indicator (Figure 1):


(6)
$$w_{j}=g_{j}/\underset{j}{\sum }g_{j}$$


### 3.2. Coupling Coordination Models

According to the weight of each indicator and the original data of each evaluation system, the comprehensive evaluation value of CT and OWB systems is calculated. The specific formula is
(7)$$U_{i}=\overset{m}{\underset{j=1}{\sum }}W_{ij}\times U_{ij}\ldots \ldots \ldots \ldots \overset{m}{\underset{j=1}{\sum }}W_{ij}=1$$

Then, the coupling and coordination degree of the CT and OWB systems is calculated.

First, according to the calculated comprehensive evaluation value, the coupling relationship (coupling degree) of the system is calculated, and the specific formula is
(8)$$C={\left(\frac{U_{1}\times U_{2}}{{\left(\frac{U_{1}+U_{2}}{n}\right)}^{2}}\right)}^{\frac{1}{2}}\;$$

In the formula, *C* is the coupling degree of the CT and OWB systems, and *U* is the comprehensive evaluation value of the two systems; the greater the coupling, the stronger the role between the two systems, and conversely, the smaller the coupling degree, the weaker the role between the two systems.

Second, the coupling coordination degree of the CT and OWB systems is calculated to illustrate their relationship as a whole, and the specific formula is
(9)$$\{\begin{array}{c} D=\sqrt[{2}]{C\times T}\\ T=aU_{1}+bU_{2} \end{array}$$

In the formula, *D* is the coupling and coordination degree of the CT and OWB systems, *T* is the comprehensive adjustment index of the two systems, and a and b are the pending coefficients of the two systems. According to the balance between the CT and OWB systems, which cannot harm the benefits of other subsystems, it is considered that the weight of the system is equal, and according to the existing research results, it is considered that the two systems have equally important roles in the development of the overall system. Therefore, the coefficient to be determined in the calculation of the coupling coordination degree of the CT and OWB systems is 1/2.

For the division of the coupling and coordination relationship between the CT and OWB systems, with reference to existing research [62,63,64,65], the 0.1 segmentation cut-off method was used to divide the two systems into three development stages of coordinated, transitional, and dysfunctional development, and 10 coordinated development levels from extreme imbalance to high-quality coordination, and each major category was divided into three specific types: lagging CT system, lagging OWB system, and synchronized CT and OWB system (Table 3).

### 3.3. Spatial Variability Analysis

The spatial characteristics of the coupling coordination of the CT and OWB systems were analyzed using the Thiel index, standard deviation, and coefficient of variation, as follows.

#### 3.3.1. Thiel Index

The Thiel index, or Thiel entropy standard, which was named by Theil (1967), uses the concept of entropy in information theory to calculate income inequality. The Thiel index is decomposable and is an important indicator to measure the characteristics of regional differences; the larger the value, the greater the difference between regions, and vice versa. Referring to existing studies [60,66], the spatial differences in the coupling and coordination of the CT and OWB systems in 12 provinces (regions) in western China were studied, and the specific formula is as follows:(10)$$T=\frac{1}{N}\overset{N}{\underset{i=1}{\sum }}\left(y_{ij}/\bar{y}\right)\ln\left(y_{ij}/\bar{y}\right)$$

In the formula, *T* is the Thiel index for the degree of difference between the CT and OWB systems as a whole; *y_ij_* and *y* represent the coupling coordination level of the *j*th year in province (region) *i* and the coupling coordination development level of the year; and *N* is the number of provinces.

#### 3.3.2. Standard Deviation and Coefficient of Variation

In order to further verify the spatial differences in the coupling and coordination of the CT and OWB systems in the 12 western provinces (regions), the calculation results of standard deviation (*S*) and coefficient of variation (*V*) were used in the relative and absolute sense.
(11)$$S=\sqrt{\overset{n}{\underset{i=1}{\sum }}\left(G_{i}-\bar{G}\right)/n}$$
(12)V=SG¯

In the formula, *G_i_* is the coupling coordination level of the *i*th province (region); n is the number of provinces (regions), which is 12; and *G* is the average coupling coordination level of the 12 provinces (regions).

## 4. Results

### 4.1. Comprehensive Evaluation of Time Series Change Analysis

The comprehensive development level of the CT and OWB systems from 2007 to 2020 was calculated by the entropy weight method. In general, the trend of the comprehensive development level of the two systems was basically consistent; except for a decline in the comprehensive development capacity of the CT system in 2020, the development speed of both systems in the remaining years was basically the same, indicating that these systems in western China showed a rapid and balanced development trend (Figure 4 and Table 4).

First, the comprehensive development capacity of the CT system fluctuated and was easily affected by special events, and its vulnerability characteristics are obvious. The comprehensive evaluation value of cultural tourism increased from 0.0611 in 2007 to 0.6012 in 2020. Except for the great impact of the coronavirus epidemic in 2020, the highest comprehensive evaluation value during the period was 0.8656 in 2019, and the average annual growth rate from 2007 to 2019 was 24.72%, showing a rapid development trend. The fastest growth rates of the comprehensive evaluation value of cultural tourism from 2007 to 2019 were found in Chongqing Municipality, Tibet Autonomous Region, Qinghai Province, Guizhou Province, and Gansu Province, with increases of 262.89, 251.45, 215.56, 202.17, and 134.25%, respectively, in 2020 compared with 2007, while slower growth rates were found in Guangxi, Yunnan, and Xinjiang, which increased by 53.95, 66.07, and 30.99%, respectively, in 2020 compared with 2007 (Figure 4 and Table 4).

Second, the comprehensive development capacity of the OWB system steadily improved. The comprehensive evaluation value of residents’ objective well-being increased from 0.0761 in 2007 to 0.8745 in 2020, with an average annual growth rate of 22.84%, and the development level of OWB continued to improve. In 2020, objective well-being in the western region was affected by the coronavirus epidemic; the comprehensive evaluation value increased by 2.93% compared with 2019, and the growth rate was significantly reduced. From 2007 to 2020, the fastest growth rates of the comprehensive value of objective well-being in different provinces (regions) were in Guizhou Province, Tibet Autonomous Region, Chongqing Municipality, Gansu Province, and Yunnan Province, with average annual growth rates of 25.17, 17.58, 16.97, 16.20, and 15.13%, respectively, during the period, while the average annual growth rate of other provinces (regions) was higher than 10%, and the 12 provinces (regions) showed a trend of rapid development (Figure 4 and Table 4).

### 4.2. Coupled Coordinated Timing Change Analysis

The coupling and coordination between the CT and OWB systems was found to fluctuate and develop and changed from the stage of high coupling and low coordination to high coupling and high coordination. That is, the coupling degree of the two systems remained above 0.9, indicating a high correlation between the two from 2007 to 2020. The coupling coordination degree increased from 0.2611 in 2007 to 0.8515 in 2020, an increase of 226.12% and an average annual increase of 9.52%, divided into three development stages: imbalanced (2007–2008), transition (2009–2011), and coordinated development (2012–2020) (Table 5).

First, in the offset stage (2007–2008), the coupling coordination degree of the CT and OWB systems increased from 0.2611 in 2007 to 0.3338 in 2008, an increase of 27.84%, and the growth rate was relatively fast. This was mainly manifested as two types, namely, moderately imbalanced cultural tourism with synchronized resident welfare, and mildly disordered cultural tourism with synchronized resident welfare, and the comprehensive development level of cultural tourism and residents’ objective well-being was low, although the two showed a high degree of correlation but relatively weak coordinated development ability. This is also related to the impact of the global financial crisis in 2008. By 2008, Sichuan Province, Xinjiang Uygur Autonomous Region, and Shaanxi Province were in the transitional development stage, which involved cultural tourism and objective welfare simultaneously. The rest of the provinces (regions) all showed dysfunctional development. Inner Mongolia showed seriously imbalanced lagging cultural tourism; Guizhou and Tibet were moderately dysfunctional with synchronous CT and OWB systems; Guangxi, Chongqing, Yunnan, and Gansu were mildly dysfunctional with synchronous CT and OWB systems; and Ningxia and Xinjiang showed moderately imbalanced lagging cultural tourism.

Second, in the transition period (2009–2011), the coupling coordination degree of the CT and OWB systems increased from 0.4292 in 2009 to 0.5249 in 2011, an increase of 22.30%, and the growth rate was relatively fast. This was mainly manifested in the transition stage, after the impact of the global financial crisis; the comprehensive development capacity of residents’ objective well-being continued to improve, while the continuous improvement of the vitality of cultural tourism was not consistent with that, and the product supply and market demand of cultural tourism were weak; thus, there was still a certain gap regarding the needs of residents’ spiritual lives. In 2011, the six provinces (regions) of Inner Mongolia, Guangxi, Guizhou, Tibet, Qinghai, and Ningxia were in the stage of dysfunctional development, among which Inner Mongolia and Tibet showed moderately dysfunctional lagging cultural tourism, Qinghai and Ningxia showed mildly dysfunctional lagging cultural tourism, Guangxi was bordered by dysfunctional cultural tourism with synchronous objective welfare, and Guizhou showed mildly dysfunctional cultural tourism and synchronous objective welfare. Chongqing, Sichuan, Yunnan, Shaanxi, Gansu, and Xinjiang were in the transitional development stage, among which Chongqing and Xinjiang were on the verge of dysfunctional lagging cultural tourism, Sichuan and Shaanxi were barely coordinated cultural tourism with synchronous objective welfare, and Yunnan and Gansu were on the verge of imbalanced cultural tourism with synchronous objective welfare.

Third, in the coordinated development stage (2012–2020), the coupling coordination degree of the CT and OWB systems increased from 0.6058 in 2012 to 0.8515 in 2020, an increase of 24.57%, with a relatively fast growth rate from 2012 to 2020. In this stage, except for the impact of the coronavirus epidemic in 2020, the development of cultural tourism showed obvious vulnerability, while the development ability of OWB continued to improve. From 2007 to 2019, the coupling and coordinated development of cultural tourism and residents’ objective well-being changed from primary coordination to high-quality coordination (2019), showing that stable cultural tourism and residents’ well-being were synchronous and complemented each other; cultural tourism promotes the development of residents’ objective well-being, and conversely, the development of OWB continuously optimizes the overall environment for cultural tourism development to ensure its sustainable development. The highest level of development at this stage was in Sichuan Province in 2019, specifically for the well-coordinated synchronous CT and OWB systems, while in 2020, Inner Mongolia, Tibet, Qinghai, and Ningxia were in the transitional development stage; only Inner Mongolia showed lagging, barely coordinated cultural tourism, and the remaining three were on the verge of imbalanced lagging cultural tourism. Guangxi, Chongqing, Sichuan, Guizhou, Yunnan, Shaanxi, Gansu, and Xinjiang were in the stage of coordinated development, among which Guangxi, Guizhou, Yunnan, Gansu, and Xinjiang showed lagging primary coordinated cultural tourism, Chongqing and Shaanxi showed lagging intermediate coordinated cultural tourism, and Sichuan showed lagging well-coordinated cultural tourism.

### 4.3. Spatial Differentiation Feature Changes of Coupling Coordinates

The Thiel index for the five northwestern provinces and five southwestern provinces in the western region was calculated. It was found that the spatial gap in the western region was getting smaller, the development levels among the 12 provinces (regions) continued to increase, and the spatial difference was greater in the southwest region than in the northwest region (see Figure 5a).

First, on the whole, the spatial differences in the western region were getting smaller. The Thiel index decreased from 0.0264 in 2007 to 0.0140 in 2020, for an overall decrease of 46.92% and an average annual decrease of 4.75%, indicating that the spatial difference between the coupling and coordination level of cultural tourism and objective welfare in the 12 western provinces (regions) was getting smaller.

Second, the spatial differences within the western region were obvious, showing a clear downward trend, and the spatial differences were greater in the southwestern region than in the northwestern region. The Thiel index for the southwestern region was 0.0386 in 2007, which was 0.0183 higher than the 0.0203 in the northwestern region, and 0.0144 in 2020, which was 0.0007 higher than the 0.0138 in the northwestern region, narrowing the gap between CT and OWB by 96.40% from 2007.

Third, the spatial differences within the western region decreased significantly, and the southwestern region decreased significantly faster than the northwestern region. From 2007 to 2020, the spatial difference between the coupling level of the CT and OWB systems in the southwestern region decreased by 62.58%, with an average annual decrease of 7.28%, while in the northwest region it decreased by 32.17%, with an average annual decrease of 2.94%, indicating that the trend of narrowing spatial difference between CT and OWB was significantly faster in the southwestern region than in the northwest region.

Calculating the standard deviation and coefficient of variation of the coupling coordination of the CT and OWB systems in western China from 2007 to 2020, it can be seen that the relative gap widened, but the absolute gap gradually narrowed (see Figure 5b).

First, the relative gap in the coupling and coordination level of the CT and OWB systems in the western region widened significantly, but the absolute gap narrowed. From 2007 to 2020, the standard deviation increased from 0.0741 to 0.1025, an increase of 38.28% and an average annual growth of 2.52%. In contrast, the coefficient of variation decreased from 0.2261 in 2007 to 0.1646 in 2020, a decrease of 27.22% and an average annual decrease of 2.41%.

Second, the relative gap and absolute gap in the northwestern and southwestern regions showed opposite development trends. In terms of the relative gap, it was greater in the southwest than in the northwest, but the expansion rate was smaller than in the northwest. The standard deviation for the northwest increased from 0.0651 in 2007 to 0.0952 in 2020, an increase of 46.40% and an average annual growth of 2.98%. The standard deviation for the southwest region increased from 0.0859 in 2007 to 0.1098 in 2020, an increase of 27.26% and an average annual growth of 1.90%. In terms of the absolute gap, it narrowed faster in the southwest than in the northwest, and the two tended to be equal by 2020. The coefficient of variation for northwest China decreased from 0.2016 in 2007 to 0.1657 in 2020, a decrease of 17.84% and an average annual decrease of 1.50%. The coefficient of variation for southwest China decreased from 0.2762 in 2007 to 0.1664 in 2020, a decrease of 39.74% and an average annual decrease of 3.82%.

## 5. Discussion

A coupling and coordination model of the CT and OWB systems was constructed, and the empirical research was carried out by taking western China as an example for the research period, and the results showed the following.

First, the comprehensive development level of the CT and OWB systems continued to improve. The western region continued to promote the optimization of the cultural tourism system, and the comprehensive evaluation value of cultural tourism increased from 0.0611 in 2007 to 0.6012 in 2020, reaching the highest value of 0.8656 in 2019. By improving the policy support system, tapping cultural tourism resources, creatively developing cultural tourism products, enhancing brand competitiveness, and broadening the path of tourist source markets, the comprehensive development capacity of cultural tourism in the western region will continue to improve. For example, the characteristics and competitiveness of attractions such as the “Little Tumbler Sister” in Shaanxi Province, Dunhuang Mogao Grottoes in Gansu Province, Lijiang Ancient City in Yunnan Province, ethnic tourism in Guizhou Province, and rural tourism in Sichuan Province will continue to improve, and the comprehensive development capacity of cultural tourism will continue to improve.

The fastest growth rate of the comprehensive evaluation value of cultural tourism from 2007 to 2019 was in Chongqing Municipality, Tibet Autonomous Region, Qinghai Province, Guizhou Province, and Gansu Province, while the growth rate was slower in Guangxi, Yunnan, and Xinjiang Provinces. The comprehensive evaluation value of residents’ objective well-being increased by 22.84% from 2007 to 2020, and the development level of objective well-being continued to improve. In 2020, the objective well-being in the western region was affected by the coronavirus epidemic; the comprehensive evaluation value increased by 2.93% compared to 2019, and the growth rate was significantly reduced. From 2007 to 2020, the fastest growth rate of the comprehensive value of objective well-being was in Guizhou Province, Tibet Autonomous Region, Chongqing Municipality, Gansu Province, and Yunnan Province, with an average annual growth rate of 25.17, 17.58, 16.97, 16.20, and 15.13%, respectively, while the average annual growth rate of other provinces (regions) was higher than 10%, and 12 provinces (regions) showed a trend of rapid development. The CT and OWB systems both showed a continuous development trend; however, “the economic development of the western region has led to the imbalance of tourism development and the further widening of the income gap between urban and rural areas, and the comprehensive benefits of rich natural and cultural tourism resources with outstanding attraction for tourists have not been fully utilized” [67], but the level of economic development and infrastructure conditions limit the sustainable development of rural tourism in the region.

Second, the ability to couple and coordinate the CT and OWB systems continued to improve. The two systems were divided into three development stages, namely, dysfunctional stage (2007–2008), transitional stage (2009–2011), and coordinated development (2012–2020), and the coupling and coordination degree increased from 0.2611 in 2007 to 0.8515 in 2020, an increase of 226.12% and an average annual increase of 9.52%. The highest level of development during the period was in Sichuan Province in 2019, specifically for well-coordinated cultural tourism with synchronous objective welfare. In 2020, Inner Mongolia, Tibet, Qinghai, and Ningxia were in the transitional development stage. Only Inner Mongolia showed lagging barely coordinated cultural tourism, and the remaining three were on the verge of imbalanced lagging cultural tourism. Guangxi, Chongqing, Sichuan, Guizhou, Yunnan, Shaanxi, Gansu, and Xinjiang Provinces were in the stage of coordinated development, among which Guangxi, Guizhou, Yunnan, Gansu, and Xinjiang showed lagging primary coordinated cultural tourism, Chongqing and Shaanxi showed lagging intermediate coordinated cultural tourism, and Sichuan showed lagging well-coordinated cultural tourism. This has a lot to do with the fact that “the western region is rich in cultural tourism resources, but the level of development is low, and it is more dependent on the overflow of factor resources from otherewhere” [68].

Third, the spatial equilibrium of the CT and OWB systems is obvious. The spatial differences in the western region are getting smaller, and the absolute gap is narrowing, but the relative gap has widened significantly. The level of coupling and coordination between the CT and OWB systems in the 12 western provinces (regions) decreased by 46.92%, with an average annual decrease of 4.75%; the coefficient of variation decreased by 27.22%, with an average annual decrease of 2.41%. The spatial difference decreased significantly within the western region but was greater in the southwestern region than in the northwestern region. The absolute gap narrowed faster in the southwest than in the northwest, and the two tended to be equal by 2020. The Thiel index for the northwestern and southwestern regions decreased by 32.17 and 62.58%, respectively, and the gap between the two narrowed by 96.40%; the standard deviation increased by 46.40% for the northwestern region and 27.26% for the southwestern region. The coefficient of variation decreased by 17.84% for the northwestern region and 39.74% for the southwest region, with an average annual decrease of 3.82%.

Fourth, the CT and OWB systems are coupled for coordinating high-quality development. Developers of cultural tourism destinations study market segmentation targets and the elasticity of supply and demand [69] from a global perspective, aiming to solve the problem of cost sharing and benefit sharing [70], realize the overall development of cultural tourism on a larger scale, and provide sustainable support to give full play to cultural tourism for objective well-being. In particular, the role of cultural tourism in meeting people’s social needs and the pursuit of happiness, and the benign operation of society [50] have become increasingly prominent, but the comprehensive development capacity of cultural tourism supply and demand in the western region lags behind objective well-being; for example, the development capacity in Guangxi, Guizhou, Yunnan, Gansu, and Xinjiang is far lower than the speed of objective welfare development and cannot meet the growing spiritual needs of people in these regions. These regions should improve their comprehensive service capability, provide rationally laid out parking facilities, and optimize the overall cultural tourism routes to improve the efficient use of facilities such as cultural centers, libraries, and museums. By enriching industrial formats, broadening financing channels, shaping more characteristic IP, and building professional talent teams, the transformation and upgrading of traditional cultural tourism carriers such as A-level attractions will be realized. By improving the policy system and optimizing the supply of high-quality cultural tourism products, we can better meet people’s needs for a spiritual life.

## 6. Conclusions

Based on coupling coordination theory, in this paper we constructed two systematic evaluation index systems for CT and OWB based on existing research and the United Nations Millennium Eco-System Assessment Report, which mainly includes 18 index systems, and divided the development of the coupling and coordination level into three stages of dysfunctional, transitional, and coordinated development, and 10 levels of high-quality coordination. The comprehensive evaluation value of cultural tourism increased from 0.0611 in 2007 to 0.6012 in 2020, and the fastest growth rate of this value from 2007 to 2019 was in Chongqing, while the growth rate was slower in Guangxi. The average annual growth rate of objective well-being from 2007 to 2020 was 22.84%, with the fastest growth rate in Guizhou Province, while the average annual growth rate of other provinces (regions) was higher than 10%, and 12 provinces (regions) showed a rapid development trend. The CT and OWB systems were in a state of transition from high coupling and low coordination to high coupling and high coordination, which were divided into three development stages, namely, imbalance, transition, and coordinated development, and the coordination degree developed steadily from moderate imbalance to good coordination over time. The spatial equilibrium of the CT and OWB systems is obvious, and the spatial difference in the western region is getting smaller, but the relative gap is significantly widening.

This paper explored the coupling and coordination relationship between cultural tourism and residents’ well-being, selecting 2007–2020 as the research period and taking western China as an example, mainly starting from the time China’s 5A-level tourist attractions were considered as important cultural tourism development carriers, and selecting the five-year plan for national economic and social development as the node. However, further optimization is needed for the selection of the index system, and different scales and types can be selected for comparative research, and the coordination level of the two systems of cultural tourism and objective welfare can be studied in depth according to the new development stage, statistical data, and research methods.

## Figures and Tables

**Figure 1 ijerph-20-00650-f001:**
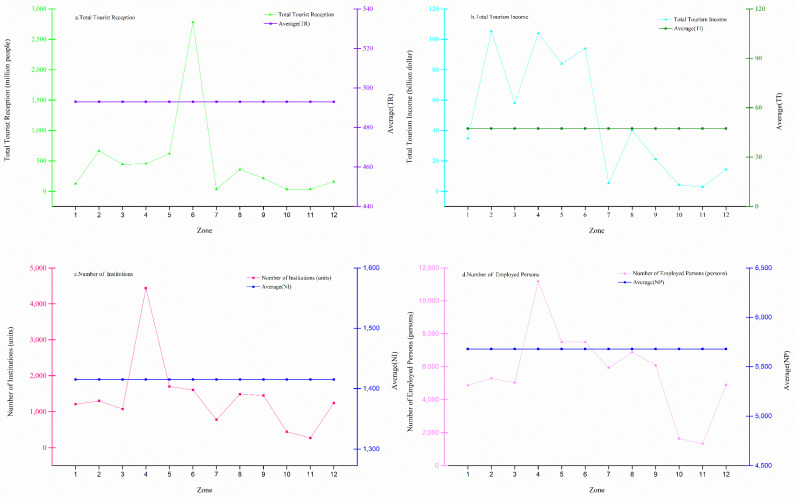
Line charts of tourist arrivals, tourism revenue, mass cultural institutions, and employment in 12 western provinces (regions) in 2020. According to information from the National Bureau of Statistics of China, USD and RMB will be converted according to their average exchange rates in 2020: 1 US dollar to 6.8974 yuan. Zones 1–12 represent Inner Mongolia Autonomous Region (1), Guangxi Zhuang Autonomous Region (2), Chongqing (3), Sichuan Province (4), Guizhou Province (5), Yunnan Province (6), Tibet Autonomous Region (7), Shaanxi Province (8), Gansu Province (9), Qinghai Province (10), Ningxia Hui Autonomous Region (11), and Xinjiang Uygur Autonomous Region (12). (**a**) Total number of tourist receptions in 12 provinces (regions) and average number in western region in 2020; TR represents total number of tourist arrivals (million people). (**b**) For 2020, total tourism revenue of 12 provinces (regions) and average income of western region; TI represents total tourism revenue (billion USD). (**c**) Total tourism revenue of 12 provinces (regions) and average income of western region in 2020; NI represents number of mass cultural institutions. (**d**) Employment at mass cultural institutions in 12 provinces (autonomous regions) in 2020; NP represents number of employed persons.

**Figure 2 ijerph-20-00650-f002:**
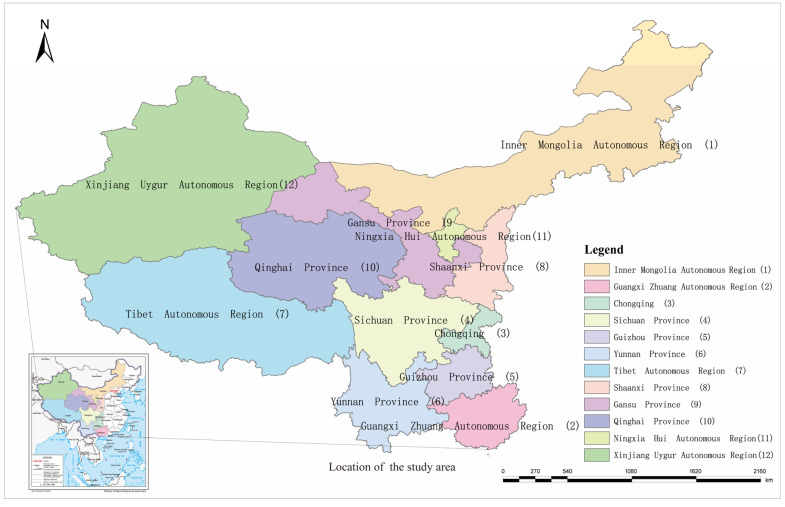
Location of the study area.

**Figure 3 ijerph-20-00650-f003:**
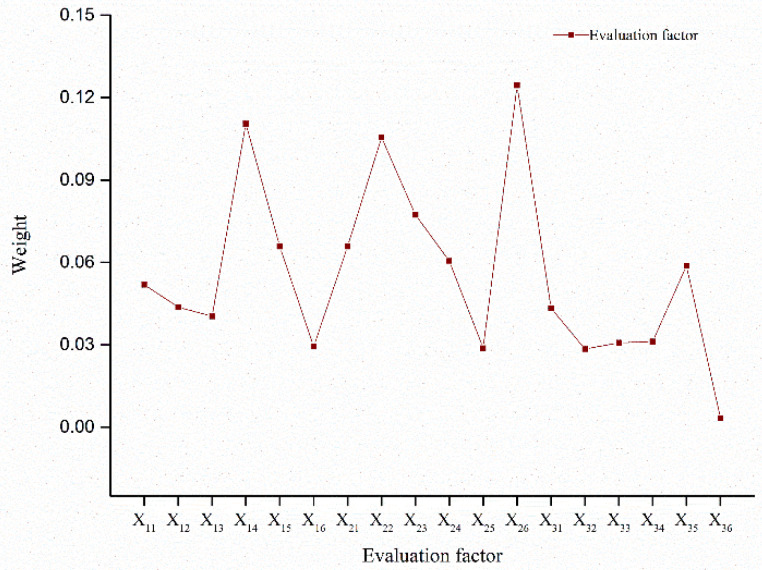
Evaluation factor index weight for CT and OWB systems. It mainly represents the weight of 18 evaluation factors such as number of 5A tourist attractions in cultural tourism and subjective well-being.

**Figure 4 ijerph-20-00650-f004:**
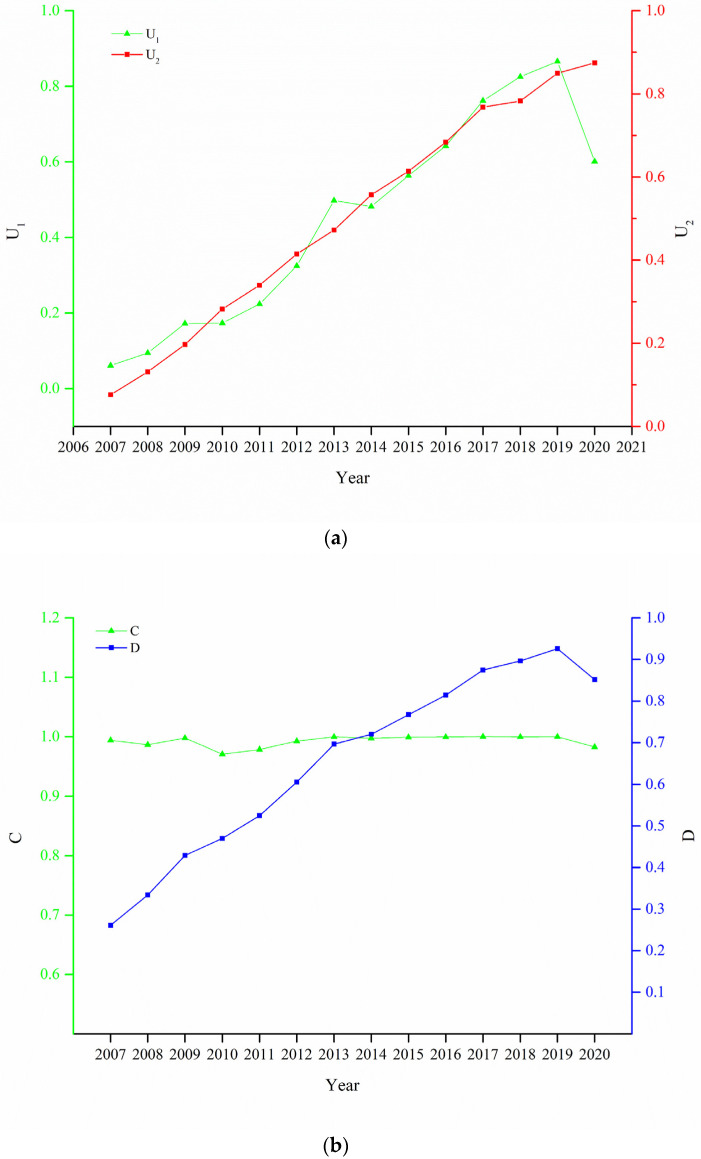
Line charts showing comprehensive evaluation value of cultural tourism and residents’ objective well-being (**a**) and coupling coordination degree (**b**). In chart on the (**a**), U_1_ is comprehensive evaluation value of objective well-being of residents, and U_2_ is comprehensive evaluation value of cultural tourism. In chart on (**b**), C is coupling degree between cultural tourism and health and welfare, and D represents coupling coordination between the two.

**Figure 5 ijerph-20-00650-f005:**
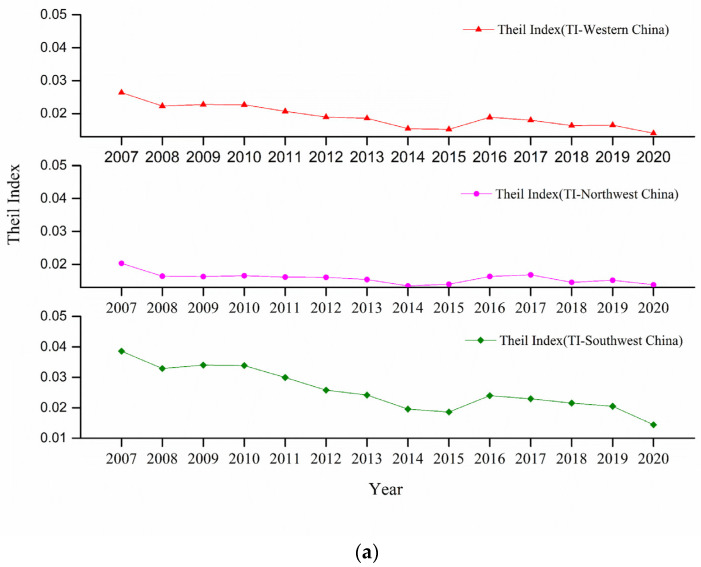
Line charts of Thiel index, standard deviation, and coefficient of variation in five western and northwestern provinces (regions) and five southwestern provinces (regions) in China. Northwest China includes Shaanxi Province, Gansu Province, Qinghai Province, Ningxia Hui Autonomous Region, and Xinjiang Uygur Autonomous Region. Southwest China includes Chongqing, Sichuan Province, Guizhou Province, Yunnan Province, and Tibet Autonomous Region. The graphs (**a**) show Thiel index for western, northwestern, and southwestern regions. The graphs (**b**) show standard deviation and coefficient of variation for the three regions.

**Table 1 ijerph-20-00650-t001:** Main evaluation index system of cultural tourism and residents’ objective well-being.

First Index	Second Index	Evaluation Factor	Factor	Unit	P/N *
CT system	Supply capacity	Number of 5A tourist attractions	X_11_	Number	+
		Number of star tourist hotels	X_12_	Number	+
		Number of travel agencies	X_13_	Number	+
		Number of artistic performance groups	X_14_	Number	+
		Number of museums	X_15_	Number	+
		Number of cultural centers	X_16_	Number	+
	Demand capability	Per capita tourism consumption of domestic tourists	X_21_	USD/person	+
		Per capita tourism consumption of inbound tourists	X_22_	USD/person	+
		Total library circulation	X_23_	10,000 person-times	+
		Total visitors to museums	X_24_	10,000 person-times	+
		Domestic art performance attendance	X_25_	10,000 person-times	+
		Cultural market institutions operating income	X_26_	USD	+
OWB system	Comprehensive consumption ability	Per capita disposable income	X_31_	USD/person	+
	Health service capability	Healthcare personnel per thousand population	X_32_	Number of persons	+
		Medical institution beds per thousand population	X_33_	Number of medical beds	+
	Environmental service ability	Per capita public green space area	X_34_	m^2^	+
	Safety assurance ability	Public security expenditure	X_35_	Billion USD	+
		Traffic accident death and injury rates	X_36_	Rate	−

* “+” indicates that the factor is a positive indicator, and “−” indicates that it is a negative indicator.

**Table 2 ijerph-20-00650-t002:** Major statistical changes in cultural tourism in 2020.

Zone	5A Tourist Attractions	Star Tourist Hotels	Number of Travel Agencies	Number of Artistic Performance Groups	Museums	Cultural Centers
Inner Mongolia Autonomous Region	6	205	1202	204	172	117
	6.19%	6.25%	13.08%	5.05%	10.54%	9.61%
Guangxi Zhuang Autonomous Region	6	444	881	78	142	116
	6.19%	13.53%	9.59%	1.93%	8.70%	9.52%
Chongqing	10	163	714	1265	105	43
	10.31%	4.97%	7.77%	31.29%	6.43%	3.53%
Sichuan Province	15	356	1336	725	258	207
	15.46%	10.85%	14.54%	17.93%	15.81%	17.00%
Guizhou Province	8	231	665	200	92	100
	8.25%	7.04%	7.24%	4.95%	5.64%	8.21%
Yunnan Province	9	415	1105	270	161	149
	9.28%	12.65%	12.02%	6.68%	9.87%	12.23%
Tibet Autonomous Region	5	165	310	87	8	81
	5.15%	5.03%	3.37%	2.15%	0.49%	6.65%
Shaanxi Province	11	325	862	591	309	117
	11.34%	9.91%	9.38%	14.62%	18.93%	9.61%
Gansu province	5	384	780	347	226	104
	5.15%	11.70%	8.49%	8.58%	13.85%	8.54%
Qinghai Province	4	207	515	122	24	50
	4.12%	6.31%	5.60%	3.02%	1.47%	4.11%
Ningxia Hui Autonomous Region	4	89	164	30	54	27
	4.12%	2.71%	1.78%	0.74%	3.31%	2.22%
Xinjiang Uygur Autonomous Region	14	297	657	124	81	107
	14.43%	9.05%	7.15%	3.07%	4.96%	8.78%
Western China	97	3282	9192	4044	1633	1219
	100%	100%	100%	100%	100%	100%

**Table 3 ijerph-20-00650-t003:** Main evaluation index system of cultural tourism and residents’ objective well-being. It mainly uses the “0.1 segmentation cut-off method” to divide the coordination level and specific types of cultural tourism and objective welfare systems.

Stage of Development	Rank of Harmony Degree	Concrete Types	
Coordinated development [0.6, 1]	Better coordination [0.9, 1] Well coordinated [0.8, 0.9) Intermediate coordination [0.7, 0.8) Lower coordination [0.6, 0.7)	Lagging CT system Lagging OWB system Synchronized CT and OWB systems	U_2_-U_1_ ≥ 0.1 U_1_-U_2_ ≥ 0.1 U_1_ < 0.1 and U_2_ < 0.1
Transitional development [0.4, 0.6)	Barely coordinated [0.5, 0.6) On the verge of imbalance [0.4, 0.5)	Lagging CT system Lagging OWB system Synchronized CT and OWB systems	U_2_-U_1_ ≥ 0.1 U_1_-U_2_ ≥ 0.1 U_1_ < 0.1 and U_2_ < 0.1
Dysfunctional development [0.0, 0.4)	Mild disorder [0.3, 0.4) Moderate outrage [0.2, 0.3) Severe dysregulation [0.1, 0.2) Extreme dysregulation [0, 0.1)	Lagging CT system Lagging OWB system Synchronized CT and OWB systems	U_2_-U_1_ ≥ 0.1 U_1_-U_2_ ≥ 0.1 U_1_ < 0.1 and U_2_ < 0.1

**Table 4 ijerph-20-00650-t004:** Increased comprehensive evaluation values of CT and OWB in 2007–2020.

2007–2020	CT System		OWB System	
	Increase in Magnitude	Average Annual Growth Rate	Increase in Magnitude	Average Annual Growth Rate
Inner Mongolia Autonomous Region	107.73%	5.78%	289.10%	11.02%
Guangxi Zhuang Autonomous Region	53.95%	3.37%	464.02%	14.23%
Chongqing	262.89%	10.42%	667.18%	16.97%
Sichuan Province	108.07%	5.80%	466.64%	14.27%
Guizhou Province	202.17%	8.88%	1751.25%	25.17%
Yunnan Province	66.07%	3.98%	524.46%	15.13%
Tibet Autonomous Region	251.45%	10.15%	721.13%	17.58%
Shaanxi Province	131.82%	6.68%	383.46%	12.89%
Gansu province	134.25%	6.77%	603.99%	16.20%
Qinghai Province	215.56%	9.24%	359.38%	12.44%
Ningxia Hui Autonomous Region	132.54%	6.71%	293.29%	11.11%
Xinjiang Uygur Autonomous Region	30.99%	2.10%	269.89%	10.59%

**Table 5 ijerph-20-00650-t005:** Coupling coordination degree and type of CT and OWB systems in western China from 2007 to 2020.

Stage of Development	Year	C	D	Specific Type
Dysfunctional development	2007	0.9940	0.2611	CT and OWB systems synchronized in moderate imbalance
	2008	0.9865	0.3338	CT and OWB systems synchronized in mild disorder
Transitional development	2009	0.9978	0.4292	CT and OWB systems synchronized on the verge of imbalance
	2010	0.9710	0.4703	Lagging CT system on the verge of imbalance
	2011	0.9786	0.5249	Lagging CT system, barely coordinated
Coordinated development	2012	0.9926	0.6058	CT and OWB systems synchronized in low coordination
	2013	0.9997	0.6964	CT and OWB systems synchronized in low coordination
	2014	0.9974	0.7199	CT and OWB systems synchronized in intermediate coordination
	2015	0.9991	0.7670	CT and OWB systems synchronized in intermediate coordination
	2016	0.9995	0.8141	CT and OWB systems synchronized and well coordinated
	2017	1.0000	0.8746	CT and OWB systems synchronized and well coordinated
	2018	0.9997	0.8966	CT and OWB systems synchronized and well coordinated
	2019	1.0000	0.9260	CT and OWB systems synchronized in better coordination
	2020	0.9827	0.8515	Lagging CT system, well coordinated

## Data Availability

The data is from *Zhongguo Wenhua Wenwu He Lvyou Tongji Nianjian* and *China’s Tertiary Industry Statistical Yearbook*.

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
