# Peer review of "Spatiotemporal Characteristics of Coupling and Coordination of Cultural Tourism and Objective Well-Being in Western China"

_ijerph, 2022, doi:10.3390/ijerph20010650_

Round 1
Reviewer 1 Report
Dear Authors, you have to think about the Reader, who for sure will be interested in your research, but will be tiered from reading so much figures and charters. Please, simplify the text. Introduce just those chartes you think realy important. Pay more attention explaining the results and the causes for differences in results for different provinces.

Author Response
- The is one extra word “zone”. And I think the athours meant that “zone names are listed sequentially”– then it is important to introduce the numbers of each of them (like: Inner Mongolia Autonomous Region (1); Guangxi Zhuang Autonomous Region (2)、Chongqing (3) etc.so as it will be easier for the reader to find the exact zone on the chart (in Fig.1).
The zone zones 1-12 in the legend represent each other: Inner Mongolia Autonomous Region 152、Guangxi Zhuang Autonomous Region、Chongqing、Sichuan Province、Guizhou Province、 153Yunnan Province、Tibet Autonomous Region、Shaanxi Province、Gansu Province、Qinghai 154Province、Ningxia Hui Autonomous Region and Xinjiang Uygur Autonomous Region.
Modified to:” Zones 1–12 in the legend represent each other: Inner Mongolia Autonomous Region(1), Guangxi Zhuang Autonomous Region(2), Chongqing(3), Sichuan Province(4), Guizhou Province(5), Yunnan Province(6), Tibet Autonomous Region(7), Shaanxi Province(8), Gansu Province(9), Qinghai Province(10), Ningxia Hui Autonomous Region(11), and Xinjiang Uygur Autonomous Region(12).”
- I guess the author meant “the data’ not “the numbers”,
The numbers of The Number of 5A tourist attractions, Number of star tourist hotels, 168 Number of travel agencies, Number of artistic performance groups, Number of museums, 169 Number of cultural centres, Per capita tourism consumption of domestic tourists, Per cap- 170 ita tourism consumption of inbound tourists, Total Library Circulation, Total visitors to 171 the museum, Art performance group domestic performance audience, Cultural market 172 operating institutions operating income are derived from the Zhongguo Wenhua Wenwu He 173 Lvyou Tongji Nianjian.
Yes, we modified to: “The numbers of 5A tourist attractions, star tourist hotels, travel agencies, artistic performance groups, museums, and cultural centers, per capita tourism consumption of domestic tourists, per capita tourism consumption of inbound tourists, total library circulation, total visitors to museums, domestic art performance attendance, and cultural market institutions operating income are derived from the Zhongguo Wenhua Wenwu He Lvyou Tongji Nianjian.”
- The Figures and Tables are not mentioned in the text. They are in a way look like text itself. But then it makes text very much complicated to understand. The Provinces are not numerated – so it makes it impossible to match the graphics and text with the name of the provinces. For Europeans, there is no understanding of these provinces (where are they, what are their differences, what makes them so different in graphic terms, etc.).
Modified and added. Figure 1 is labeled in the appropriate position in the text,for example with a total of 5915.54 million tourist trips and tourism revenue of USD 568.09 billion (see a and b in Figure 1 for details). As of 2020, the number of mass cultural institutions in the western region was 16,981, employing 68,145 people, with an average of 4 employees at each cultural institution. (see c and d in Figure 1 for details) Similarly, place the icons and text in the article accordingly.
- The map of the area (Chapter 2.1.) is obviously needed to indicate the location of each of the 12 Provinces.
Modified and added “Figure 2. Location of the study area.”
- The table with some cultural object and some characteristics of the Provinces will be of special importance to let the Reader better understanding / feeling the area and its background for cultural tourism development from one side and challenges the provinces face.
Modified and added “Table 2. Major statistical changes in cultural tourism in 2020.” Again, increased “The western region is located in the western region of China and plays an important role in China's cultural tourism, ethnic tourism, eco-tourism and rural tourism. As of the end of 2020, there were 4,625 A-grade tourist attractions, accounting for 34.85% of the national A-level tourist attractions. From the perspective of natural resources, the western region accounts for 72% of the country's land area, and the terrain falls from the roof of the world to the low-altitude plain, and the climate is distributed vertically. Since the policy support of reform and development and the large-scale development of the western region, the development of tourism in the western region has also shown a rapid growth trend, and tourism hotspots have emerged, such as Dunhuang Mogao Grottoes, Qinghai Chaka Salt Lake, Qin Shihuang Terracotta Army, Shapotou and other important tourist scenic spots "cultural and tourism integration", "cultural tourism" and other hot spots have become hot words. The development of the cultural tourism industry in the western region mainly relies on important cultural routes such as the traditional Silk Road and the Tibet-Yi corridor to attract domestic and foreign tourists, and the comprehensive role of tourism in economic development, social construction and foreign exchanges has been continuously brought into play”
- To make more citing for the sentences where there are more than 2 sources are cited as I think they are worth to be mentioned separately:
Ex.: According to the particularity of China's economic development, cultural tourism 133 according to the existing research[46,47], and the particularity of industrial development, 134 from the supply and demand of cultural tourism to build a holistic system, its Chinese 135 chapter selects 5A-level tourist attractions, star-rated hotels, travel agencies, museums, 136 cultural centers and art performance groups as an important part of the cultural tourism 137 product system, which plays an important role in the quality of life of residents and the 138 perception of tourists' happiness[48,49,50,51].
Modified to: According to the particularitiesy of China’s economic development, cultural tourism according to the existing cultural tourism research [53,], and the particularitiesy of industrial development [54], in order to build a holistic cultural tourism system based on-from the supply and demand of cultural tourism to build a holistic system, its Chinese chapter selects 5A-level tourist attractions, star-rated hotels, travel agencies, museums, cultural centers and art performance groups should be considered as an important parts of the cultural tourism product system, which playings an integral important role in the quality of life of residents and tourists’ the perception of tourists’ happiness [55–58].
Reviewer 2 Report
This research is very good, but it is not methodologically made to be a scientific article, but only a professional article. The problem is in the methodology because the purpose of the research is not stated in the introduction, the goal is not stated, and the research hypothesis is also not stated. However, the article cannot be a scientific article.
Author Response
This research is very good, but it is not methodologically made to be a scientific article, but only a professional article. The problem is in the methodology because the purpose of the research is not stated in the introduction, the goal is not stated, and the research hypothesis is also not stated. However, the article cannot be a scientific article.
We modified and ,we Add purpose to the introduction.
“However, at present, the development of cultural tourism is facing problems such as abundant resources but relatively backward economic and social development[12], and the development opportunities of cultural tourism integration under the background of digital economy development[13], but China's cultural tourism development is character-ized by abundant resources and strong economic strength[14]. How to realize the cou-pling and coordinated development of the two systems of cultural tourism and objective welfare has become an important driving force for the sustainable development of region-al economy, and also an important measure for the continuous improvement of the com-prehensive benefits of cultural tourism. Therefore, taking the coupling and coordinated development of the two systems of cultural tourism and objective well-being as the start-ing point, this paper constructs a coupling and coordination model of the two systems, and conducts empirical analysis in western China.
First, relying on the actual situation of China's development and industrial charac-teristics, build a comprehensive evaluation index system of two systems of cultural tour-ism and objective welfare, including a total of 2 systems, 6 secondary indicators and 18 evaluation factors, and then build a coupling and coordination model of the two systems of cultural tourism and objective welfare, which is divided into 3 development stages and 10 coordination levels. Taking western China as an example, this paper provides a typical case of the coupling and coordinated development law and its characteristics of the two systems of cultural tourism and objective welfare in China, and enriches the relevant the-ories of the interactive development of cultural tourism and objective welfare.
Second, through the overall construction of the cultural tourism product system of regional destinations, build a more comfortable, safe and convenient objective welfare system, and provide China's typical cases and sustainable development paths for the coupling and coordination of high-quality development of cultural tourism and objective welfare.“
Reviewer 3 Report
The article is well presented, but the key terms like Objective Wellbeing, tourism and more specifically cultural tourism needs to be introduced better. What are the definitions you follow? Why? Especially the definition what you understand as a tourist is very relevant for your calculations and results. Please explain better.
Author Response
The article is well presented, but the key terms like Objective Wellbeing, tourism and more specifically cultural tourism needs to be introduced better. What are the definitions you follow? Why? Especially the definition what you understand as a tourist is very relevant for your calculations and results. Please explain better.
We modified and ,we Add Objective Wellbeing, and cultural tourism needs to be introduced in the introduction.“In 1985, the World Tourism Organization (UNWTO) defined the conceptual meaning of cultural tourism from both broad and narrow levels, and in 1991, the European Associ-ation for Tourism and Leisure Education (ATLAS) defined cultural tourism as "all non-profit activities in the location of cultural resources such as ancient ruins, artistic and cultural performances, and artistic interpretations in order to obtain or meet their own cultural needs". The 1999 International Charter on Cultural Tourism elaborates on the meaning of cultural tourism, which regards culture itself and the environment as the core of cultural tourism, and the cultural environment includes not only historical and cultural landscape resources, but also local residents' customs, lifestyles and even natural land-scape resources of tourist destinations. Tourism and culture go hand in hand and com-plement each other [15], cultural communication plays an important role in the develop-ment of tourism economy[16], likewise, the process of tourism presents diverse cultural needs[17]. With the development of economy and society, the concept of cultural tourism is also constantly changing, tourism has become a necessity of people's life, and more at-tention is paid to the experience of regional culture. Therefore, the article understands cultural tourism as a way of people's modern life. Objective well-being mainly represents people's happiness and spiritual happiness in life[18], and both of them are highly corre-lated in spiritual life, because this paper takes western China as an example to analyze the coupling and coordination relationship between cultural tourism and objective well-being to further promote the sustainable development of the region.”
Round 2
Reviewer 1 Report
Dear Authors, thank you for taking into concideration my comments. THe paper now looks much better.
Reviewer 3 Report
The authors well addressed and incorporated the reviewer's comments. The article is now ready to be published.